# Comprehensive Testing of Sulfate Erosion Damage of Concrete Structures and Analysis of Silane Coating Protection Effect

**DOI:** 10.3390/s22207991

**Published:** 2022-10-20

**Authors:** Dunwen Liu, Yinghua Jian, Yu Tang, Kunpeng Cao, Wanmao Zhang, Haofei Chen, Chun Gong

**Affiliations:** School of Resources and Safety Engineering, Central South University, Changsha 410083, China

**Keywords:** sulfate erosion, ultrasonic speed measurement, CT scanning, silane protection, damage analysis

## Abstract

In order to study the protection performance of silane coating on in-service concrete structures in a sulfate environment, we collect concrete samples in the field to simulate the concrete erosion process by accelerated erosion with wetting–drying cycles. We place the samples into protected, exposed and control groups corresponding to a corrosive environment with silane protection, corrosive environment without protection and general environment for three different service conditions. A combination of ultrasonic velocimetry, CT (Computed Tomography) scan imaging, NMR (Nuclear Magnetic Resonance) pore structure analysis, strength testing and other methods are used to analyze the strength, ultrasonic wave velocity, pore structure and other characteristics of the specimens during sulfate erosion. Based on the test results, the protective effect of silane coating on concrete structures under sulfate attack is quantitatively analyzed, and an index for judging the damage rate of specimens is proposed to quantitatively analyze the protective effect of silane coating. The research results show that the damage of the concrete structure under silane protection in a sulfate-attack environment can be reduced by more than 50%; its integrity damage index and strength damage index are easily affected by the location of local defects, which leads to a decrease in the protection efficiency of the surface silane coating.

## 1. Introduction

In the last three decades, bolstered by large-scale infrastructure construction, a huge amount of concrete structures in service have existed in China. In the western region of China, there are a large number of salt lakes, caves and deposits rich in sulfate components which are in contact with concrete bodies using water as a medium. This causes more serious damage to the durability of concrete structures through combined internal–external erosion and physical–chemical corrosion [1,2,3,4,5]. As an early line built in China in the southwestern karst region, the products of physical salt erosion, ettringite sulfate erosion and thaumasite sulfate erosion were collected in a large number of Chengkun Railway tunnels during the investigation of its service [6]. Subsequent model experimental analysis showed that the erosion of tunnel-lining concrete by sulfates is the main cause of cracking and spalling of tunnel linings [7]. A similar situation occurred at Bapanxia Dam [8] and Lijiaxia Dam [9] in the northwestern region. Sulfate minerals in the underlying rock and soil bodies came in contact with the concrete via groundwater transport, causing severe damage to the concrete structure in the projects. Some mines have a high content of sulfate-like substances in their wastewater produced during the mining process. It corrodes the concrete support structures in the mine shafts, which causes serious safety hazards [10,11]. From the current cases, some areas have high sulfate content in groundwater. Tunnels with a 50-year design life often start to show serious damage after 3–5 years of use [12,13]. It is easy to see that the erosive effect of sulfates seriously affects the durability performance of concrete structures. Knowing how to carry out protection work is urgently needed to ensure safe and stable structural performance.

Usually, sulfate enters the concrete interior by penetration using water as a medium. It leads to structural deterioration through mechanical or chemical interaction with the cement paste [14,15,16], which in turn reduces or removes load-bearing capacity. Factors such as engineering environment, material properties and concentration of erosive materials can all influence the erosion process. In actual projects, structures often have several types of erosion. Therefore, sulfate corrosion of concrete in engineering structures is often the result of the combined effect of multiple corrosion modes.

To counteract sulfate damage to concrete structures, two methods are usually used to enhance the erosion resistance of concrete. One method is to adjust the ratio of concrete. By adding additives and improving cement varieties, the structure and environment inside concrete can be adjusted so as to inhibit the production of corrosion products [17]. This approach is economical and feasible in the construction phase. However, for the operational phase, mass concrete replacement not only seriously increases maintenance costs but also seriously increases safety hazards. Another treatment, namely surface protection technology, forms a hydrophobic layer on the concrete surface by means of corresponding materials. In turn, harmful ions are prevented from entering the concrete interior with the help of water penetration [18,19]. Silane, as a common hydrophobic protective material, is ideal for constructing a surface-protection barrier for concrete in sulfate-rich conditions. Concrete impregnated with silane usually possesses a high hydrophobicity [20]. Even foamed concrete with high porosity has better results [21,22]. The solution using impregnated silane-protected concrete is easy to implement and has achieved good results in the maintenance of some projects where sulfate erosion hazards have occurred [23,24]. It is worth noting that some engineering investigations have shown that silane coatings still provide some protection after many years of use [25]. However, the protective effect of harsh-environment coatings may decrease with time [26,27]. Therefore, it is necessary to establish a scientific analysis and evaluation method for the protection effect to select a reasonable protection method.

The prediction of the service-life of concrete impregnated with silane by means of field statistics combined with probabilistic models can be a feasible way to judge the protection effect [25,28]. However, the relevant factors vary from project to project, and generalized model predictions face difficulty in obtaining results that are fairly close to reality. For concrete structures in operation, nondestructive testing methods are used to test and obtain key characteristic parameters. The analysis of concrete durability without destroying its integrity has unique technical advantages. For example, ultrasonic velocimetry methods can be tested to obtain the acoustic wave velocity values within the concrete and analyze the variation patterns of the acoustic wave velocity within the concrete. In turn, the degree of damage to the existing concrete relative to its initial state is judged [29,30,31,32]. This method has been widely used in studying and testing concrete properties. However, a corrosive environment may lead to heterogeneity brought about by the structure, and it becomes difficult to characterize the local structural deterioration from the overall wave velocity data. It should be considered in combination with other structural tests as a complement [5]. CT imaging technology is used as a technical tool that can visually analyze the internal structural characteristics of concrete [33,34,35,36,37]. It can visually reflect the internal structural state of the specimen and reveal the derivation process of defects. However, its data are mostly pictorial descriptions, which are difficult to correlate directly with the mechanical properties of the sample. Combining the respective advantages of ultrasonic inspection and CT imaging techniques enables a more comprehensive characterization of the damage to the specimen. NMR is a test method to determine the surface area-to-volume ratio in porous media by measuring the relaxation time of fluid molecules within the pores of a specimen under conditions of an applied external magnetic field [38]. Specific porosity parameters and pore distribution can be determined. The NMR technique is used to enrich comprehensive concrete damage detection and assessment, and to verify the accuracy of the combination of ultrasonic testing and CT imaging techniques [39].

Existing studies have shown that silane protection has a positive effect on the prevention of sulfate attack as an effective protective measure. However, silane-coating protection mainly depends on the hydrophobic layer generated on the concrete surface, and the surface integrity directly determines its protection effect. According to the requirements of JTJ275-2019, the thickness of a silane protection layer is mostly 2–4 mm. In the actual service process, if the protection layer is damaged by external collision, stress cracks or other factors, local protection failure may occur. Referring to the study of service tunnels in the literature [2], in the parts of the tunnel with high wind velocity, its rapid wetting–drying cycles rendered it prone to the problem of concrete surface damage caused by crystallization damage, which can also lead to local area protection failure; studies in the literature [19,20] also show that the performance of protective coatings is not constant. Therefore, in order to objectively assess the health status of concrete under coating protection in actual projects, there is an urgent need to establish an analytical evaluation method that takes into account the overall performance and local characteristics. In this paper, we used a combination of ultrasonic velocimetry and CT scanning, together with NMR and strength testing, to conduct nondestructive testing of concrete specimens from a river-crossing tunnel-tube sheet structure subjected to sulfate erosion processes. In order to analyze and evaluate the effectiveness and effect of concrete protection measures under sulfate erosion conditions, this paper uses a combination of ultrasonic velocimetry and CT internal structure scanning test method to conduct nondestructive testing on concrete specimens from a river crossing tunnel tube sheet structure subjected to sulfate erosion process, characterize the damage degree of the specimens by test indexes, and use the concrete specimen strength test as a reference and test to compare the protected specimens with the unprotected specimens, and then establish a method to analyze and evaluate the effectiveness and effect of protection measures under sulfate erosion conditions. The strength test of the concrete specimens was used as a reference and test [40,41], to compare the protected specimens with the unprotected specimens and analyze the protection effect of the silane coating, so as to establish a method to analyze and evaluate the protection effect under sulfate erosion and to judge the effectiveness and effect of the protection measures taken in the actual project by the results of the comparison test.

## 2. Materials and Methods

### 2.1. Specimen Preparation and Experimental Design

#### 2.1.1. Specimen Preparation

The specimens selected for this experiment are from a river-crossing tube sheet in service. The design strength grade is C50, the 28-day compressive strength is 61.6 MPa and the water–cement ratio is 0.4, according to the relevant research [42,43] and the ratio parameters provided on site; see Table 1.

To prepare the specimens, the poured pipe pieces were first broken using a crushing hammer to extract the reinforcement inside the pipe pieces. A concrete core sample with a diameter of 50 mm was extracted using core-drilling equipment. Then, the core sample was processed into a 50 mm × 100 mm cylindrical specimen. The specimens were prepared for testing, with an age of 1150 d and an original compressive strength of 75 MPa.

The original tunnel used isooctyltrioxethylsilane silane to spray a protective coating on the surface of the tube sheet before it was put into use. The experimental specimens used in this paper restore the use of silane materials on site, and the designed amount of silane for concrete in the reference project is 450 g/m^2^. The surface area of the specimens used in this paper is about 0.01963 m^2^, and the corresponding design amount is about 8.8357 g. Considering the accuracy required for actual operation and measurement, the actual amount of silane used in this paper is 8.9 g for each specimen. The specimens were divided into a protected group, exposed group and controlled group. The protected group was coated with the designated amount of silane before the experiment, and the amount used per single specimen was 8.9 g. The exposed group and the controlled group were left untreated.

#### 2.1.2. Experimental Cycle Design

The wetting–drying experimental maintenance cycle consists of an immersion phase (wet phase) and a drying phase (dry phase). The experimental immersion phase simulates the erosion process of the tunnel-tube sheets by immersing the test blocks in the erosion solution of the corresponding composition. During the immersion period, the test blocks of the protected and exposed groups were placed upright in a PVC box with a lid containing a 10% mass fraction of sodium sulfate solution. The controlled group was placed in the same style of PVC box with water. The top of the specimen was placed in the PVC box with the same water, the distance between the top of the specimen and the liquid level in the box was not less than 3 cm and the solution in the box was changed once every 30 days. The set maintenance temperature was 20 ± 0.5 °C, and the relative humidity was 95 ± 0.5%. The specimens entered the drying stage after the specified soaking time. The specimens were taken out from the erosion liquid box, the residual liquid on their surface wiped off and then placed in a blast-drying box. The specimens were dried at 60 °C for 6 h and placed in a cool and ventilated place for 2 h, then put back into the original erosion solution box for the next wetting–drying cycle. Three groups of specimens started the cycle at the same time, and the number of cycles for each group of specimens was 180 times. Preparation and curing of specimens are shown in Figure 1.

The specimen grouping and quantity design are shown in Table 2 below.

### 2.2. Specimen Testing

In this paper, three types of methods are combined to analyze the changes in the properties of concrete specimens during rapid wetting–drying cycles of erosion. Ultrasonic testing was used to analyze the overall condition of the concrete. The structural changes to the specimens were analyzed using CT scanning and NMR testing. Strength tests were used to compare the strength changes of the specimens under different conditions. The overall test protocol is shown in Figure 2.

#### 2.2.1. Ultrasonic Testing

A non-metallic ultrasonic tester was used to test the longitudinal wave velocity on the side of concrete specimens at a sampling rate of 0.05 μs. After every ten wetting–drying cycles, a test was conducted on the specimens of the protected group, the exposed group and the controlled group. The test specimens were divided into four test planes at equal intervals of 20 mm from top to bottom. Nine measurement lines are laid out at equal intervals at 20° angle for each measurement surface. The longitudinal wave velocity of each specimen was tested at 36 points in total, and the test situation is shown in Figure 2. We used an HS-CS1H ultrasonic parameter tester made by Xiangtan Tianhong Electronics Research Institute; the frequency of the arc probe is 1 MHz.

#### 2.2.2. CT Test

The specimens’ structure was analyzed using an industrial CT system. We use the model Y.CT.Modular X-CT scanning equipment, with continuous spiral scanning mode, tube voltage 225/450 KV, imaging plate size 400 mm × 400 mm, imaging plate pixel dot size 0.2 mm, detectable diameter range 0~800 mm and maximum penetration thickness 60 mm Fe. The specimens were scanned in high precision slices for the protected and exposed groups at the end of 60, 120 and 180 wetting–drying cycles. The minimum scanning resolution was 5.3 μm, and the number of slices scanned for a single specimen was 1600.

#### 2.2.3. NMR Test

After the CT tests were completed for the protected and exposed groups, we used a Suzhou Newmark AniMR-150 NMR instrument to perform NMR tests on the specimens before and after the erosion experiment to obtain pore data. The magnetic field strength of the test system is 0.3 ± 0.05 T, the peak output is not less than 300 W and the linear distortion is less than 0.5%. The specimens were soaked in water for 48 h to saturate the specimens before the test was started.

#### 2.2.4. Strength Test

The specimens of the exposed group and the specimens of the protected group that completed the NMR test were dried at 60 °C to the state before immersion. The specimens of the exposed group, protected group and controlled group were uniaxially loaded at 0.5 MPa/s using a servo material-testing machine, and the peak load when the specimen diameter *d* is damaged was recorded. The uniaxial compressive strength can be calculated according to the following formula.
$$\sigma =\frac{F}{\left(\frac{d^{2}}{4}\pi \right)}$$

Finally, the overall structure of this paper is shown in Figure 3.

## 3. Results

### 3.1. Experimental Results

(1) Appearance

Figure 4 shows the appearance of the specimens in the exposed group compared with the specimens in the protected group after 180 days. The specimens in the exposed group showed significant cracks on the surface, while the specimens in the protected group showed a more complete surface.

The surface of the exposed group specimens showed a white powdery substance at some locations on the surface before cleaning. Its composition was tested by XRD (X-ray Diffraction) and found to be mainly sodium sulfate, mostly distributed at the joints between the aggregate and the cement paste. After cleaning the surface, it was found that significant cracks were produced in the area where the white powder appeared. This indicates that these cracks are mainly due to the physical salt erosion caused by the heating of the sodium sulfate erosion solution inside the specimen during the drying process.

(2) Ultrasonic wave speed

Figure 5a reflects the average wave velocity of the measurement points of the three groups of specimens, which show different trends. The wave velocity of the exposed group specimens showed a more obvious trend of increasing and then decreasing. The mean wave velocity of specimens in the protected group also showed a similar trend, but the change was not significant. The average wave velocity of the controlled group showed a back-and-forth fluctuation, but the change was small.

Figure 5breflects the variation of the mean wave velocity at the measurement points for the three groups of specimens. The most significant variation is found in the exposed group, where the variation of velocity in a period fluctuates within the range of 10%. The wave velocity variation for specimens in the protected group was about half of that in the exposed group. In contrast, the controlled group showed essentially no change, with its maximum velocity change not exceeding 2%.

The specimens in this paper were made directly from in-service structural concrete, which has a large structural variability. This is also reflected in the variation of the wave velocity during the test period. Figure 6 shows the variation of single-point wave velocity at some measurement points of the three groups of specimens, and Figure 7 shows the variation of wave velocity variation rate at the corresponding measurement points.

Three typical patterns of variation at different measurement points of the exposed group specimens are shown in Figure 6a. Measurement point No. 7 is mainly composed of cement paste on both sides, and its deterioration form is typical of the combined action of PSA (physical salt attack) and SA (sulfate attack). The wave velocity variation also shows a clear trend of increasing and then decreasing. Measurement point No. 15’s contact location is mainly the aggregate cut surface. Since sulfate does not chemically interact with the aggregate, the PSA erosion of the aggregate is also relatively weak. Therefore, this point is the most stable point of wave velocity among the three types of points. However, the wave velocity still varies to some extent due to the internal expansion of the fracture in locations adjacent to it. Measurement point No. 35 increases in the early stage of erosion, and the velocity decreases significantly after reaching the peak. This type of measurement point is mostly at the junction position of aggregate and cement paste. After a period of wetting–drying cycles, the crystalline precipitation of sulfate crystals took the lead in destroying the joint surface of the aggregate and cement paste, resulting in cracks and a significant decrease in wave velocity.

From the rate-of-change of wave velocity in Figure 7c, the most stable point, i.e., No. 12, showed no more than 2.5% change in the original wave velocity, which was basically unchanged. Point No. 7 decreased by about 5% throughout the course of the test, and point No. 35 exceeded 15%. This indicates that the expansion of cracks at this measurement point has greatly damaged the integrity of the concrete.

Differing from the exposed group test blocks, the protected group test blocks basically showed two trends of variation. Three test points were selected in Figure 5b, among which No. 5 has a connection position between aggregate and cement paste on the surface. Point No. 10 is dominated by cement paste, but there are original air holes on the surface. Point No. 21 is a surface dominated by aggregate. It can be seen from the graph that the velocity variation of measurement points No. 5 and No. 21 is small. This indicates that the erosion of the concrete by sulfate is inhibited after the application of silane protection. However, at the same time, the wave velocity of measurement point No. 10 showed a certain degree of variation, and the trend was similar to that of the exposed group specimen measurement points. This paper speculates that due to the existence of primary defects, setting the protective coating by brushing may lead to insufficient thickness of the protective layer in the local area, thus weakening the protective effect.

Figure 6c and Figure 7c selected two types of measurement points in the controlled group, one with a single aggregate or mortar surface at point 15, and one at the material interface at point 23. the original wave velocity variation during the experiment for these two types of measurement points was lower than that of the exposed and protected groups, indicating that the wetting–drying cycle experiments in the clear water environment caused basically no damage to the specimens.

(3) Structure

It can be seen in Figure 8 that the development of pores in the exposed group specimens is more significant during the erosion process, and there is a certain degree of increase in the number of small pores on the outside of the specimens. The number of small pores on the outside of the specimens in the protected group, on the other hand, showed a significant decrease during the erosion process. The difference between the two is that the sulfate component in the erosion solution filled some of the native pores during the subsequent erosion process after the application of the silane protective coating. However, due to the protective layer, the specimen surface was maintained in a relatively good condition, which inhibited the generation of subsequent pores and fissures.

Figure 9 reflects the overall pore-structure change and the respective pore size composition of the two groups of specimens during the erosion process. From the overall porosity, both show a trend of first decreasing and then increasing, and the structure of the protected group changes less and is more stable. It can be seen in Figure 9b that, after 180 days of erosion, the number of small-size pores in the exposed group specimens decreased and the number of large-size pores increased to some extent. This shows that the specimens as a whole maintain the process of transformation from small pores to large pores, and the small pores are still being generated continuously. In contrast, the volume number of large pore size decreased after the erosion of the specimens in the protected group. This stems from the fact that the apparent large pores of the specimens are filled with eroded sulfate. The number of very small pores increased to some extent, indicating that defects may have developed at a defect location that is not yet visible to the naked eye.

(4) Strength

The uniaxial compressive strengths of each group are shown in Table 3. As seen in the table, the strengths of the controlled group were basically the same as the original specimens, while the uniaxial compressive strengths of both the exposed and protected groups showed significant reductions. Compared with the original specimens, the strength of the exposed group was reduced by about 27% and that of the protected group reduced by about 17%. The results show that the silane coating can effectively reduce the erosion of concrete due to sulfate under the experimental design of wetting–drying cycle erosion conditions, but it cannot completely avoid the deterioration of concrete caused by sulfate erosion.

### 3.2. Damage Metric

#### 3.2.1. Structural Damage Analysis

Unlike other erosion-damage processes, sulfate erosion of concrete is a process of strengthening before damage. Therefore, within the erosion process, the area where the concrete is in contact with the erosion medium will have some of its pores filled first. Along with the subsequent erosion process, the concrete structure begins to deteriorate, and new pore structures are formed one after another.

If the porosity p of concrete during erosion is monitored, the p values mostly show a state of first decreasing and then increasing. If the erosion process is carried out several times in succession, the sequence of porosity test values is $\left\{p_{i}\right\}$, defined as:(1)$$p^{\prime }=min\left\{p_{i}\right\},$$
where $p^{\prime }$ characterizes the pore volume inside the specimen that is not affected by the erosion component during the filling phase of erosion. For the porosity $p_{i}$ obtained at a subsequent test, the volume $\Delta p_{i}$ of new pores generated by erosion is:(2)$$\Delta p_{i}=p_{i}-p^{\prime },$$

Since the newly generated pores must be caused by the material damage resulting from erosion, then the equivalent volume $\Delta p_{i}$ at this point represents pores which have been damaged by the material. According to the basic definition of damage, the percentage $d_{p}$ of its pore damage is [43]:(3)$$d_{p}=\Delta p_{i}/p_{0},$$
where $p_{0}$ is the initial porosity of the material.

Based on the above idea, we can calculate the $d_{p}$ of the exposed group and the protected group, respectively, after 180 days. The corresponding parameters are shown in Table 4.

The calculations in Table 3 show that the structural damage in the protected group is only half of that in the exposed group. This demonstrates the remarkable effect of silane protective coating on the protection of the surface.

#### 3.2.2. Strength Damage Analysis

In this paper, based on the uniaxial compressive strength results of the specimens, the strength-loss of the material is calculated by the following equation:(4)$$d_{s}=1-\frac{f_{c}}{f_{0}},$$
where $f_{c}$ is the compressive strength of the material after erosion and $f_{0}$ is the original compressive strength of the material.

Based on the strength test results above, we can conclude that the strength damage for the exposed group is 0.274 and for the protected group is 0.174.

#### 3.2.3. Ultrasonic Wave Speed Damage Metric

In order to improve the accuracy of ultrasonic wave velocity testing, this paper divides the specimen into four equally spaced measurement surfaces $H_{1},\;H_{2},\;H_{3},\;H_{4}$, with each surface containing a total of nine measurement lines.

Dynamic modulus $E_{d}$ of general specimen elasticity, density ρ, longitudinal wave velocity v and Poisson’s ratio μ satisfies the following relationship [44]:(5)$$E_{d}=\frac{\left(1+\mu \right)\left(1-2\mu \right)\rho v^{2}}{1-\mu },$$

Let $k=\frac{\left(1\;+\;\mu \right)\left(1\;-\;2\mu \right)\rho }{1\;-\;\mu }$; then, the above equation simplifies to:(6)$$E_{d}=kv^{2},$$

In the concrete sulfate erosion experiment, the composition k of each parameter is a constant value. The dynamic elastic modulus of the specimen is linearly related to the longitudinal wave velocity at the measurement point. The change in the longitudinal wave velocity of the specimen can be used to characterize the change in the kinematic modulus and indirectly responds to its strength characteristics.

As a result, the measurement surface is divided by the center of the measurement line, and the longitudinal wave velocity measured by the measurement line represents the area where the measurement line is located. The equivalent kinetic mode $E_{d}^{\prime }$ is defined as:(7)$$E_{d}^{\prime }=\sum kw_{i}v_{i}^{2},$$
where $v_{i}$ is the measured line wave speed, and $w_{i}$ is the proportion of the measurement line area occupying the overall area. In this paper, nine lines are evenly arranged in each measurement plane to divide the measurement plane, so the $w_{i}$ value is 1/9.

Using the equivalent dynamic modulus as the damage factor, the initial equivalent dynamic modulus of the specimen is $E_{d0}^{\prime }$. The damage degree of the measured surface can be calculated as follows:(8)$$d_{f}=\frac{E_{d0}^{\prime }-E_{d}^{\prime }}{E_{d0}^{\prime }}$$

Damage degree $d_{f}$ reflects the damage variation of the material strength parameters in the test area centered on the ultrasonic velocity measurement surface, and is a more-comprehensive response to the overall performance of the measured area. Table 5 shows the results of damage calculation for each side of the exposed and protected groups.

From the calculation results in Table 5, it can be seen that the actual calculated damage degree of each measurement surface varies greatly. The damage factors calculated for some of the measured surfaces were small and approximated no damage. For the $H_{4}$ measured surface in the exposed group and $H_{3}$ and $H_{4}$ measured surfaces in the protected group, the damage factors calculated were more consistent with the damage factors obtained from the uniaxial strength calculations. Based on the same structure of the four measured surfaces, it can be assumed that the most severely damaged surface will be the first to suffer damage when the load is applied. Therefore, it is more reasonable to use the most severely damaged area as the overall damage characterization of the specimen.

Therefore, this paper considers that the damage degree of the specimen should be taken as the maximum value of the damage degree of the four measurement surfaces.

### 3.3. Protection Efficiency Analysis

When comparing and verifying the protection effect of specimens in the engineering field, due to the variability between specimens, the protection effect can be judged by comparing the damage parameters d and $d^{\prime }$ of protected and unprotected specimens in the same time period. Then, for a certain test value, we define the protection efficiency value E, i.e.:(9)$$E=1-\frac{d}{d^{\prime }},$$

The E value represents the mitigation effect of protective measures on structural damage, and a larger E value indicates a better protective effect.

From the results in Table 6, it is clear that silane protection has the highest protection efficiency in terms of structural damage and lower efficiency in terms of strength and wave speed damage. As an overall protection method, silane protection can indeed play an effective role in isolating the concrete structure from sulfate environmental exposure so that the structure as a whole remains in a more stable state. However, the strength and wave velocity damage calculation model selected in this paper is susceptible to changes in sound velocity and strength at the weak points of protection. Therefore, to achieve a better protection effect, silane protection of the specimens should have the uniformity of the coating settings ensured, with attention paid to the protection of structural weaknesses.

However, in the actual experiments, the specimens prepared in this paper are from actual structures. There are some differences in the homogeneity of the prepared specimens.

## 4. Discussion

The results obtained from the experiments showed that the overall structure of the concrete was effectively protected by the silane coating. During the accelerated erosion tests conducted, the concrete specimen exterior and the concrete pore structure were maintained in a more stable state. The use of silane protection is undoubtedly positive and effective in safeguarding the health of concrete structures.

However, the results of the intensity and wave velocity tests were compared between the protected and exposed groups. It is easy to find that the wave velocity damage and strength damage of the silane protection specimens are smaller than that of the exposed group specimens. However, the protection efficiency in terms of strength damage and wave velocity damage is much less than that in terms of structural protection. The reason for this is that more attention was paid to the weak areas when considering the wave velocity and strength damage of the specimens. Since sulfate erosion is a result of a combination of physical damage and chemical reaction, when other areas are better protected, corrosion products at the weak point may accumulate rapidly, thus further amplifying the corrosion deterioration caused by local damage. In this paper, the raw materials used in the preparation of the style samples are from actual structures, and there are some differences in their uniformity when prepared as small-size samples. Some of the raw defects lead to a situation when the protective coating is set with silane where it is easy to miss parts and undercoat. In addition, the good water-barrier effect of silane, contrary to expectation, makes the parts with micro-defects on the surface after protection more quickly destroyed by crystallization.

In this paper, the corresponding industry specifications and similar studies were referred to when designing the test procedure [45,46]. A scheme of accelerated erosion by wetting–drying cycles was designed, but there are still some differences compared to the actual situation. For example, a higher concentration of erosion solution was used in the erosion experiments, which amplified the concrete damage brought about by PSA to some extent. At the same time, the organosilane selected for the test as a protective coating provides a good water barrier but insufficient thermal stability. At the beginning of the test conducted, a drying temperature of 60 °C was used to heat the test blocks. There was slight damage to the external coating, which indirectly weakened the protection effect. However, the current guideline in our specifications for testing the performance of concrete against sulfate for durability uses a higher drying temperature. Therefore, there is a need for further research on and development of the methods for testing the sulfate-resistance of concrete specimens under protective measures.

At present, the work on durability protection of concrete structures under conditions of sulfate erosion is still in the exploratory stage, and many structures are still in the early stage of service, lacking corresponding research accumulation. In this paper, we try to simulate the erosion process in an actual environment by accelerated erosion under the premise of field sampling. At the same time, a diversified detection system is proposed according to the key indicators of concrete erosion deterioration processes. Of course, there is still development of the corresponding detection techniques used in this paper and the subsequent data processing and analysis techniques. Through waveform recognition in ultrasonic velocimetry and image-recognition processing technology in CT inspection, the optimization of these technologies can help further the accurate evaluation of damage results. At the same time, the protection efficiency index based on damage factors proposed in this paper is still an index-pointing judging method. It is necessary to develop a systematic evaluation and analysis method to form a comprehensive judgment on the effectiveness of protection measures.

## 5. Conclusions

Various testing methods were used to judge the damage of concrete within a sulfate erosion environment. The main research results include:

1. An ultrasonic test method with multi-point determination of sound velocity was proposed to determine the characteristics of sound velocity variation during erosion deterioration. The test results demonstrate that the field-prepared specimens showed significant differences in wave velocity variation during the erosion deterioration process according to the material differences at the interface of the measurement points.

2. Sulfate erosion-acceleration experiments were carried out with actual engineering samples. The acoustic, structural and strength degradation patterns of the protected group and the exposed group under the influence of sulfate erosion were compared. It is shown that the specimens in the protected group still maintain a high overall structural integrity in the erosion test. However, there was also a more-significant loss of their compressive strength.

3. A critical index of protection efficiency based on the variation of the damage factor was proposed. The protective measures adopted in the subject tunnels were analyzed by using the critical index for coated silane protection. This type of measure has excellent effects in protecting the overall structure of the specimen. However, it is not effective for local damage or weak areas. The integrity and uniformity of the protection layer should be ensured as much as possible when used, and the existing defects should be repaired in time.

## Figures and Tables

**Figure 1 sensors-22-07991-f001:**
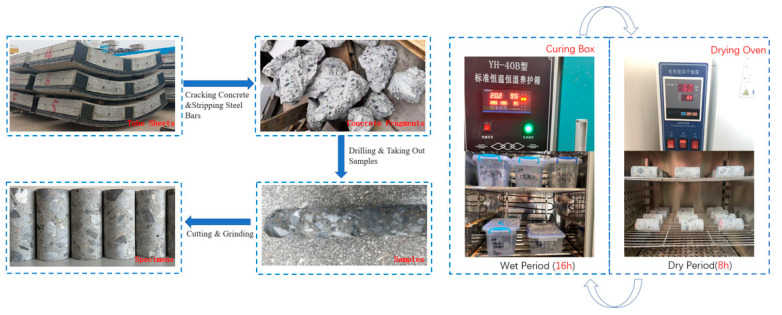
Sample preparation and test.

**Figure 2 sensors-22-07991-f002:**
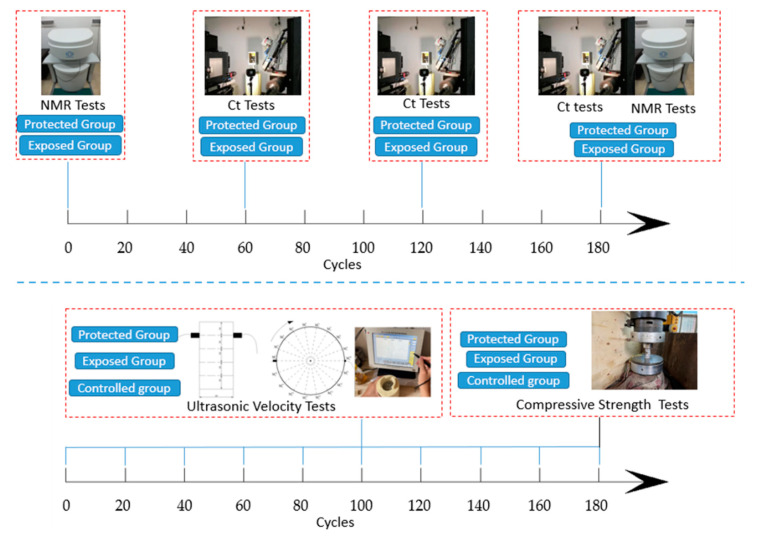
Overall test protocol of the experiment.

**Figure 3 sensors-22-07991-f003:**
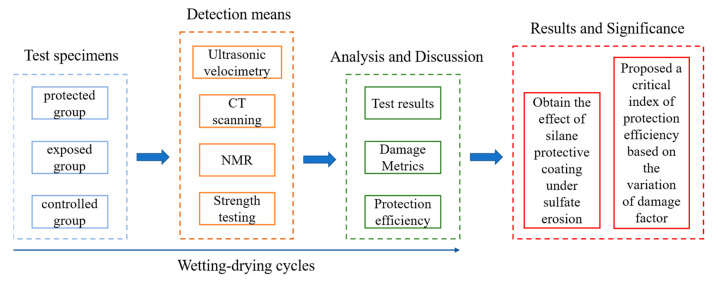
The overall structure of the paper.

**Figure 4 sensors-22-07991-f004:**
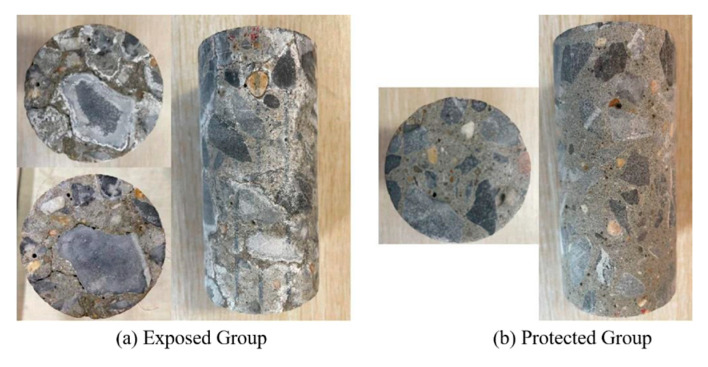
Changes in specimen appearance.

**Figure 5 sensors-22-07991-f005:**
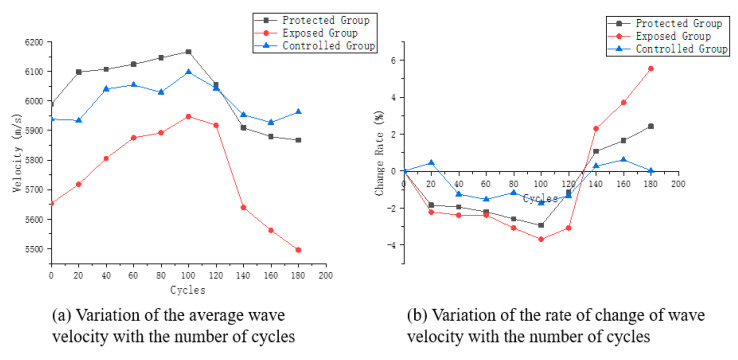
Wave velocity characteristic curve.

**Figure 6 sensors-22-07991-f006:**
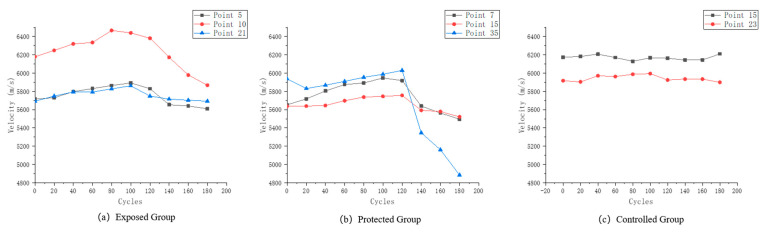
(**a**) Wave velocity variation of the characteristic measurement points of the exposed group; (**b**) Wave velocity variation of the characteristic measurement points of the protected group; (**c**) Wave velocity variation of the characteristic measurement points of the controlled group.

**Figure 7 sensors-22-07991-f007:**
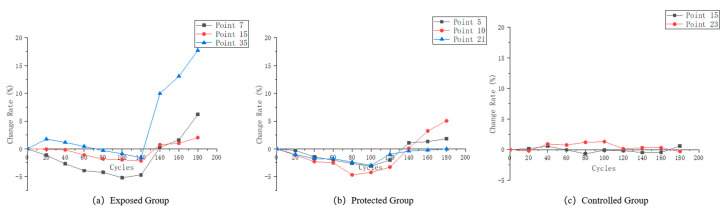
(**a**) Wave velocity variation rate of the characteristic measurement points of the exposed group; (**b**) Wave velocity variation rate of the characteristic measurement points of the protected group; (**c**) Wave velocity variation rate of the characteristic measurement points of the controlled group.

**Figure 8 sensors-22-07991-f008:**
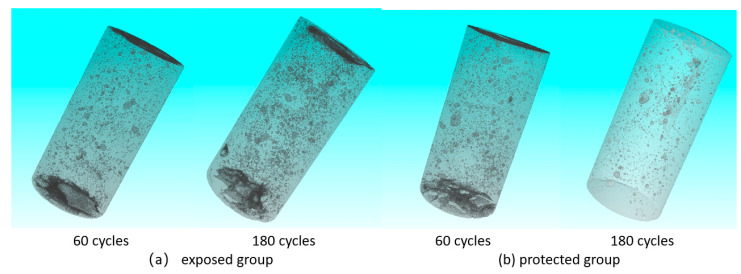
(**a**) Three-dimensional pore-structure reconstruction of specimens in the exposed group; (**b**) Three-dimensional pore-structure reconstruction of specimens in the protected group.

**Figure 9 sensors-22-07991-f009:**
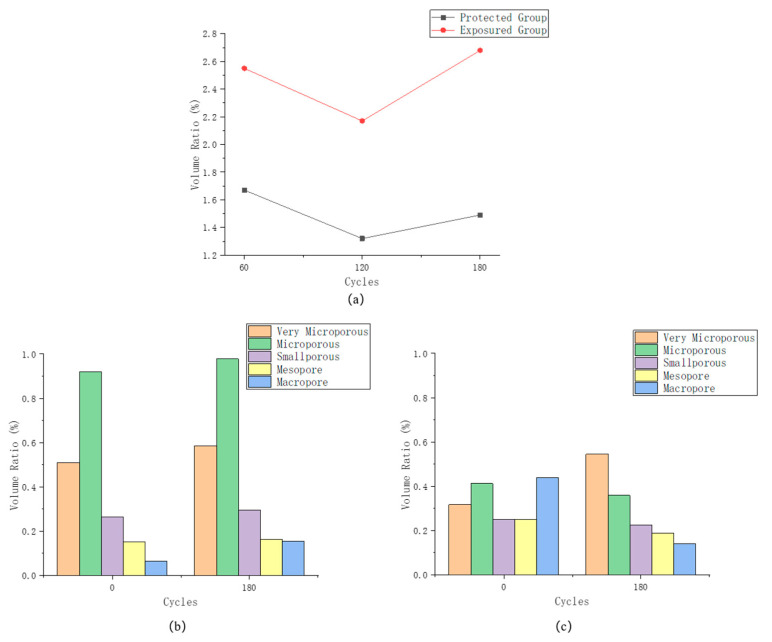
(**a**) Pore variation of specimens; (**b**) Pore composition of exposed group; (**c**) Pore composition of specimens of protected group.

**Table 1 sensors-22-07991-t001:** Concrete proportioning parameters (kg/m^3^).

Cement	Water	Sand	Gravel	Fly Ash	Expansion Agent	Water Reducer
410	165	711	1066	61	32.8	4.1

**Table 2 sensors-22-07991-t002:** Specimen grouping.

Specimens (Number)	Controlled Group (1)	Exposed Group (1)	Protected Group (1)
**Environment**	Clear water	10% Na_2_SO_4_	10% Na_2_SO_4_, Silane coating

**Table 3 sensors-22-07991-t003:** Uniaxial compressive strength of specimens.

Specimens	Exposed Group	Protected Group	Controlled Group
Failure load (KN)	104.58	118.79	140.42
Diameter (mm)	49.44	49.40	49.42
Uniaxial compressive strength (MPa)	54.48	61.98	73.21

**Table 4 sensors-22-07991-t004:** Pore damage calculation parameters.

Specimens	$p_{0}$	$p^{\prime }$	Δp	$d_{p}$
Exposed Group	2.55	2.17	0.51	0.200
Protected Group	1.67	1.32	0.17	0.102

**Table 5 sensors-22-07991-t005:** Measured surface damage calculation results.

Specimens	$d_{f1}$	$d_{f2}$	$d_{f3}$	$d_{f4}$
Exposed Group	0.070	0.035	0.090	0.227
Protected Group	0.023	0.060	0.169	0.150

**Table 6 sensors-22-07991-t006:** Protection efficiency calculation results.

Damage Factor	$d_{p}$	$d_{s}$	$d_{f}^{\prime }$
*E*	0.49	0.365	0.255

## Data Availability

The data that support the findings of this study are available upon request from the authors.

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
