# Peer review of "Comprehensive Testing of Sulfate Erosion Damage of Concrete Structures and Analysis of Silane Coating Protection Effect"

_sensors, 2022, doi:10.3390/s22207991_

Round 1

Reviewer 1 Report

In the article, the examination of concrete specimens under sulfate erosion is presented. The properties of unprotected and protected specimens (by silane coatings) are compared using four measurement methods, namely: ultrasonic velocimetry, CT scan imaging, NMR pore structure analysis, and strength testing. In the article both: the advantages and disadvantages of silane coatings application are shown.

The article has proper structure and scientific character.

In the Introduction the main problem of investigation is formulated and legitimated using 37 literature items, which were published between 2001 and 2022. However, some of the items are cited in wider groups e.g positions [1-5] (citation in line 35), without detailed comments. Please comment on the necessity of using the article with reference number [4]. In my opinion, the citation context is not proper.

In the introduction, two of four measurement techniques used in the presented examination were described. The omitted strength test for concrete specimens can be considered a basic examination. However, NMR testing for concrete should be, in my opinion, describe in more detail.

Comments to 'Materials and Methods' section:

How many specimens were prepared? The article does not specify the number of samples.

The sentence: ”The protected group was coated with the designed amount of silane before the experiment, and the amount of single specimen was 8.9g.” (line 123) is, in my opinion, unclear and should be reformulated.

Figure 2 suggests, that the Ultrasonic Velocity Tests were made only one time (after 100 cycles), but in the description in the text can be found, that it was made 'After every ten wet and dry cycles' (line 156). The drawing should match the description.

There is no information about used equipment.

Comments to 'Results' section:

Lines 203-208: How was the variation of the mean wave velocity calculated? The data on the right part of Figure 4 do not match the chart on the left side.

Line 209: Due to the lack of information about the number of specimens the interpretation of data collected in the diagrams in Figure 5 is difficult. If I correctly understand on each specimen were 36 measuring points (for wave velocity measurements). Data presented on the diagrams are for one, the same specimen or come from several samples? Are they averaged in some way?

Lack of information about the number of specimens makes difficulties in the interpretation of the other results. Eg. in Table 2 only one value of uniaxial comprehensive strength for each group is shown. I suppose, that it is a mean value, but I think, that for proper interpretation the other statistical parameters are important (like even if standard deviation).

As shown in Figures 4 and 5 the wave velocity after 180 dry-wet cycles decreased in every examined case (except the controlled group). Taking that into consideration the df parameters, calculated according to the (8) and presented in Table 4 should be, in my opinion, with a '-' sign (lines 340-348)

Editorial remarks:

Line 80: Double brackets – there is: [25][28]; should be [25, 28]

Line 91, 149, 187, 221: There are the acronyms used without previous explanation (line 91: CT imaging technology, line 149: NMR testing, line 187: XRD, line 221: PSA and SA)

Line 127: Unnecessary repetition of a word: ‘cycle’

Line 143: Figure 1 contains many pictures, which are not described. There is a lack of reference to Figure 1 in the text.

Line 179, 193 the subtitles are directly above the drawing (without text), it is confusing - are they parts of text or drawings?

Line 181: There are references to a) and b) in the caption under Figure 3, but in the figure, the letters do not exist.

Line 195: There are references to a) and b) in the caption under Figure 4, but in the figure, the letters do not exist.

Line 210: There are references to a) b), c), and d) in the caption under Figure 5, but in the figure, the letters do not exist.

Line 265: There are references to a) b) and c) in the caption under Figure 7, but in the figure, the letters do not exist.

Line 310: The table title starts with a small letter

Line 312: Wrong reference to the table. There is 'in Table 2' and should be 'in Table 3'

Author Response

dear reviewers,

we have uploaded the response as an attachment, please check the attachment.

Reviewer 2 Report

There are some weaknesses through the manuscript which need improvement. Therefore, the submitted manuscript cannot be accepted for publication in this form, but it has a chance of acceptance after a major revision. My comments and suggestions are as follows:

1- Abstract gives information on the main feature of the performed study, but some details about the conducted tests must be added. However, a concise abstract is needed.

2- Authors must clarify necessity of the performed research. Aims and objectives of the study, and also differences with the previous review papers must be clearly mentioned.

3- The literature study must be enriched. For instance, authors can read and refer to the recently published relevant papers: (a) https://doi.org/10.1016/j.apples.2021.100043 (b) https://doi.org/10.1016/j.jobe.2022.105176

4- It would be beneficial to the reader if a figure showing the overall structure of the paper to enhance the readability of the paper.

5- Appropriate reference for Table 1 is required.

6- The main reference of each formula must be cited. All figures must be depicted in a high quality. Scale bar must be added in some figures.

7- Why this particular mixture is considered? Authors can read and refer to relevant studies, such as https://doi.org/10.1140/epjst/e2018-00057-1 and https://doi.org/10.1016/j.cemconres.2018.04.007

8- Authors must explain a practical application where the concrete structural element experience wet and dry conditions considered in this study.

9- Why this particular number of cycle is considered? Is there any applications/practical usage?

10- Is there any reference (standard) for the conducted experiments? (subsection 2.2.2 and 2.2.3, and etc). They should be mentioned in details.

11- What is the crack behavior in examined specimens?

12- How uniaxial compressive strength is determined? The main formulas and details of calculations must be added in the revised version. (only numbers in Table 2 is not enough).

13- Standard deviation in the obtained results must be discussed.

14- In its language layer, the manuscript should be considered for English language editing. There are sentences which have to be rewritten.

15- The conclusion must be more than just a summary of the manuscript. List of references must be updated based on the proposed papers. Please provide all changes by red color in the revised version.

Author Response

dear reviewers:

We have uploaded the response as an attachment, please check the attachment.

Reviewer 3 Report

1.      In line 47, are you trying to say, “Tunnels with 50-year design life …”, instead of “50-year service life…”?

2.      Not sure the meaning of “The analysis of concrete durability has a research base and technical advantages.” in line 83. Please rewrite it.

3.      Please keep effectiveness or effect in line 98: they two are kind of repeating. Also, it is recommended to do a proofread of the manuscript to minimize grammar related issues.

4.      In line 122, it says “The specimens were divided into protected group, exposed group and controlled group.”. However, the Figures 5-7 in “3. Results” only shows the results from protected group and exposed group. Why?

Author Response

(The authors gave the same response as above.)
